# Conflicting Nongenomic Effects of Progesterone in the Myometrium of Pregnant Rats

**DOI:** 10.3390/ijms22042154

**Published:** 2021-02-22

**Authors:** Katsuhiko Yasuda, Aya Yoshida, Hidetaka Okada

**Affiliations:** Department of Obstetrics and Gynecology, Kansai Medical University, 2-5-1 Shinmachi, Hirakata, Osaka 573-1010, Japan; yoshiaya@hirakata.kmu.ac.jp (A.Y.); hokada@hirakata.kmu.ac.jp (H.O.)

**Keywords:** conflicting nongenomic effects, pregnant myometrium, progesterone, voltage-dependent calcium channel

## Abstract

Recently, it has been suggested that progesterone affects the contractile activity of pregnant myometrium via nongenomic pathways; therefore, we aimed to clarify whether progesterone causes and/or inhibits pregnant myometrial contractions via nongenomic pathways. Our in vitro experiments using myometrial strips obtained from rats at 20 days of gestation revealed that progesterone caused myometrial contractions in a concentration- and time-dependent manner at concentrations up to 5 × 10^−7^ M; however, this effect decreased at concentrations higher than 5 × 10^−5^ M. Similarly, progesterone enhanced oxytocin-induced contractions up to 5 × 10^−7^ M and inhibited contractions at concentrations higher than 5 × 10^−5^ M. Conversely, progesterone did not enhance high-KCl-induced contractions but inhibited contractions in a concentration- and time-dependent manner at concentrations higher than 5 × 10^−7^ M. We also found that RU486 did not affect progesterone-induced contractions or the progesterone-induced inhibition of high-KCl-induced contractions; however, progesterone-induced contractions were blocked by calcium-free phosphate saline solution, verapamil, and nifedipine. In addition, FPL64176, an activator of L-type voltage-dependent calcium channels, enhanced high-KCl-induced contractions and rescued the decrease in high-KCl-induced contractions caused by progesterone. Together, these results suggest that progesterone exerts conflicting nongenomic effects on the contractions of pregnant myometrium via putative L-type voltage-dependent calcium channels.

## 1. Introduction

Steroid hormones play many important roles in living organisms and are involved in various physiological processes, such as growth, development, anabolism, catabolism, and reproduction. The steroid hormones estrogen and progesterone are involved in reproductive organ growth and development, ovulation, fertilization, implantation, preservation of pregnancy, labor, and nursing. In the myometrium, estrogen plays an important role in increasing the uterine contractile response, whereas progesterone counteracts this effect [1,2,3,4,5]. Consequently, progesterone reduces the rate of preterm birth by inhibiting premature uterine contractions caused by uterotonic agents, including oxytocin or prostaglandin F_2α_ [6,7], and its use is recommended by published guidelines to prevent preterm birth [8,9,10].

During parturition, many factors, including oxytocin, prostaglandin, cytokines, proteases, calcium ion (Ca^2+^) channels, and contractile proteins, are involved in labor pain and participate in a common pathway that affects myometrial contractions. In smooth muscles such as the myometrium, increased intracellular Ca^2+^ levels are required for contractions induced by factors, including oxytocin, prostaglandin, and mechanical stimuli, whereas voltage-dependent calcium channels (VDCCs) and receptor-operated calcium channels (ROCCs) play key roles in increasing intracellular Ca^2+^ levels during uterine contractions [11,12]. Previously, progesterone was thought to exert specific effects on target organs via genomic pathways through the nuclear progesterone receptor (nPR). More specifically, after combining with progesterone, nPRs undergo a conformational change and dimerization, leading to translocation to the nucleus where progesterone acts as a ligand-activated transcription factor. Progesterone/nPR complexes can also modulate the activities of Src tyrosine kinase and some mitogen-activated protein kinases (MAPKs) in the cytoplasm; the MAPKs are translocated to the nucleus, where they affect gene expression. Despite these genomic pathways, most recent evidence suggests that progesterone acts via nongenomic pathways [13,14,15,16,17,18,19,20,21]. 

Some steroid hormones, including androgen and progestins, have been shown to inhibit myometrial contractions in nonpregnant rats in a nongenomic manner [13,14,15]. We previously reported that dydrogesterone, a retroprogesterone derivative with a reversed three-dimensional structure, could nongenomically inhibit myometrial contractions in pregnant rats and pregnant women [16]. In addition, it has been suggested that progesterone inhibits the contractile activity of the vascular smooth muscle by decreasing the intracellular Ca^2+^ concentration via nongenomic pathways involving the membrane progesterone receptor (mPR)-inhibitory G protein [17]. However, progesterone/mPRs coupled with the inhibitory G protein have been reported to decrease cAMP levels and increase myosin light-chain (MLC) phosphorylation to facilitate contractions in pregnant myometrium [18]. Similarly, progesterone exerts a nongenomic stimulatory effect on contractions in a pregnant myometrium [19,20,21]; however, the role of mPRs remains unclear. 

Although progesterone’s role in decreasing uterine contractions via genomic pathways has been studied extensively [22,23], few studies have investigated whether steroid hormones, including progesterone, affect the contractile activity of pregnant myometrium via nongenomic pathways. In this study, we examined the effects of steroid hormones on myometrial contractions via nongenomic pathways involving Ca^2+^ channels using myometrial strips prepared from the uterine tissues of pregnant rats. 

## 2. Results

### 2.1. Effects of Natural Steroid Hormones on Resting Myometrium

First, we evaluated the possible stimulatory effects of steroid hormones, including progesterone, on the contractile activity of the resting myometrium. In nonpregnant women, the serum concentration of progesterone is known to be 10^−7^ M in the secretory phase; however, this level increases to 10^−6^–10^−5^ M during pregnancy. In rats, the concentration of progesterone is reported to increase to 5 × 10^−7^ M during pregnancy [24]. We evaluated the effect of sequentially increasing progesterone levels in physiological saline solution (PSS; >10^−7^ M) on the contractile activity by measuring the peak area of contraction, representing frequency, tension (amplitude), and contractile time (Appendix A). The peak area of high-KCl-induced contractions was used as a reference, and the contractile activity was evaluated using the relative ratio of the peak area (RRPA; steroid hormone-induced peak area/high-KCl-induced peak area) as the absolute values of the peak area varied largely between the myometrial strips obtained from different rats.

At a concentration of 5 × 10^−7^ M, progesterone caused rhythmic myometrial contractions, and this effect increased in a concentration-dependent manner up to 5 × 10^−6^ M (Figure 1A); however, the tension appeared to decrease slightly at 10^−5^ M, whereas the contractions decreased considerably at 5 × 10^−5^ M and disappeared altogether at 10^−4^ M. Other steroid hormones, such as 17β-estradiol, testosterone, cortisol, and aldosterone, did not cause notable contractions. As shown in Figure 1B and Appendix A, the RRPA in the progesterone group increased in a concentration-dependent manner from 5 × 10^−7^ M to 5 × 10^−6^ M and was still high at 10^−5^ M; however, the RRPA decreased at 5 × 10^−5^ M and became zero at 10^−4^ M. Significant increases were observed between the progesterone and vehicle groups at the various concentrations (*p* < 0.05), whereas no significant differences were found in the RRPA of the 17β-estradiol, testosterone, cortisol, and aldosterone groups (*n* = 6, mean ± SD).

Next, we examined the effects of progesterone on contractions when added in a single dose. As shown in Figure 1C, progesterone caused rhythmic contractions with a lag at a concentration of 5 × 10^−7^ M; however, the lag disappeared, and contractions increased in a time-dependent manner at concentrations >10^−6^ M before decreasing gradually at concentrations >10^−5^ M. At 5 × 10^−5^ M and 10^−4^ M, contractions occurred sporadically for five min and then disappeared 10 min after the addition. As shown in Figure 1D and Appendix A, the RRPA increased in a time-dependent manner in the progesterone group (*n* = 6; mean ± SD) at 5 × 10^−7^ M. Conversely, the RRPA in the vehicle group (*n* = 6, mean ± SD) was negligible (Appendix A). A significant difference was observed between the progesterone and vehicle groups at 10–15 and 15–20 min. The significant and rapid increase in the RRPA of the progesterone group was more obvious at 10^−6^ M and 5 × 10^−6^ M, respectively. Moreover, the RRPA at 10^−5^ M increased significantly during the first 10 min but decreased during the next 10 min. The RRPA at 5 × 10^−5^ M and 10^−4^ M increased rapidly and significantly at 0–5 min but decreased to low levels at 5–10 min and did not increase again. We found two contractile patterns depending on the concentration of progesterone, with a time-dependent increase observed at a low concentration and a transient increase at a high concentration.

### 2.2. Effects of Progesterone on Oxytocin- and High-KCl-Induced Myometrial Contractions

It has been reported that dydrogesterone inhibits oxytocin- and high-KCl-induced contractions via nongenomic pathways [14], and progesterone can inhibit oxytocin-induced contractions at the nanomolar level by interfering with the binding between oxytocin and its receptor [25]. We examined whether progesterone inhibited oxytocin- or high-KCl-induced contractions. Unexpectedly, we found that progesterone increased the contractile time of each peak in the oxytocin (100 μunit/mL)-induced contractions up to a concentration of 5 × 10^−6^ M (Figure 2A), and the contractile time of each peak remained long at 10^−5^ M; the contractions were inhibited completely at >5 × 10^−5^ M. As shown in Figure 2B and Appendix A, the RRPA of oxytocin-induced contractions in the progesterone group was significantly higher than that in the vehicle group from 5 × 10^−7^ M to 10^−5^ M and significantly lower at concentrations >5 × 10^−5^ M (*n* = 6, mean ± SD).

Progesterone did not enhance the high-KCl-induced contractions but inhibited the contractions in a concentration-dependent manner at a concentration >5 × 10^−7^ M and completely at the concentration >5 × 10^−5^ M (Figure 2C). After the sixth medium exchange, the response to high KCl was recovered to previous levels; however, the vehicle did not affect the high-KCl-induced contraction, and the response to a high-KCl concentration was recovered immediately after the exchange. The RRPA of the high-KCl-induced contractions was significantly lower in the progesterone group (*n* = 6, mean ± SD) than in the vehicle group (*n* = 6, mean ± SD) at the concentration >5 × 10^−7^ M (Figure 2D and Appendix A) and decreased in a concentration-dependent manner.

### 2.3. RU486 Does Not Alter the Effects of Progesterone on Contractile Activity

To confirm whether the effects of progesterone were independent of genomic pathways involving nPR, we examined whether a pretreatment with RU486, a progesterone antagonist for nPR, for 60 min blocked the effect of progesterone on contractile activity in the resting myometrium and during high-KCl-induced contractions. Interestingly, a RU486 pretreatment did not inhibit rapid progesterone (5 × 10^−6^ M)-induced contractions, early progesterone (5 × 10^−5^ M)-induced contractions, or the subsequent contraction inhibitions (Figure 3A). As shown in Figure 3B and Appendix A, the RRPAs of the progesterone (5 × 10^−6^ M)-induced contractions were not significantly different between the vehicle (*n* = 6, mean ± SD) and RU486 (*n* = 6, mean ± SD) pretreatment groups. Similarly, the RRPAs of the progesterone (5 × 10^−5^ M)-induced contractions were not significantly different between the vehicle (*n* = 6, mean ± SD) and RU486 (*n* = 6, mean ± SD) pretreatment groups. The inhibitory effect of progesterone (5 × 10^−6^ M) on the high-KCl-induced contractions was not blocked by the RU486 pretreatment for 60 min (5 × 10^−7^ M; Figure 3C). As shown in Figure 3D and Appendix A, the RRPAs of high-KCl-induced contractions were not different between the vehicle (*n* = 6, mean ± SD) and RU486 (*n* = 6, mean ± SD) pretreatment groups. Thus, RU486 did not affect progesterone-induced contractions in the resting myometrium or the progesterone-induced inhibition of high-KCl-induced myometrial contractions.

### 2.4. Calcium-Free PSS and VDCC Inhibitors Alter the Stimulatory Effect of Progesterone in the Resting Myometrium, Whereas FPL64176 Alters the Inhibitory Effect of Progesterone on High-KCl-Induced Myometrial Contractions

VDCCs are known to play important roles in increasing intracellular Ca^2+^ levels, which are required for myometrial contractions. L-type calcium channels are an important type of VDCC, which also includes N-, P/Q-, R-, and T-type channels. We investigated the role of calcium influx in the stimulation of the resting myometrium by progesterone using calcium-free PSS, verapamil (a VDCC inhibitor), and nifedipine (an L-type VDCC inhibitor). Interestingly, progesterone (5 × 10^−6^ M) did not cause rhythmic contractions in calcium-free PSS (Figure 4A). As shown in Figure 4B and Appendix A, the RRPA in the progesterone group increased in the presence of PSS (*n* = 6, mean ± SD); however, it reached a value of zero at every five-min interval in the presence of calcium-free PSS. Significant differences were observed at every five-min interval between the two groups. As shown in Figure 4C, a 15-min pretreatment with nifedipine (10^−7^ M) and verapamil (5 × 10^−7^ M) completely inhibited progesterone (5 × 10^−6^ M)-induced contractions, unlike that with the vehicle. As shown in Figure 4D and Appendix A, the RRPAs in the vehicle pretreatment group (*n* = 6, mean ± SD) increased in a time-dependent manner and were significantly greater than those in both the nifedipine and verapamil pretreatment groups (*n* = 6, mean ± SD) at every five-min interval.

Finally, we examined whether FPL64176, an activator of L-type VDCCs [26,27], influenced the inhibitory effects of progesterone on high-KCl-induced contractions. As shown in Figure 4E, high-KCl-induced contractions decreased for 20 min after the addition of progesterone (5 × 10^−6^ M), increased after the addition of FPL64176 (10^−7^ M), and returned to previous levels 20 min later but continued to decrease 20 min after the vehicle was added. High-KCl-induced contractions did not change for 40 min following the addition of the vehicle alone; however, FPL64276 increased the contractions for 40 min. As shown in Figure 4F and Appendix A, the RRPAs of high-KCl-induced contractions in the progesterone + FPL64176 group (*n* = 6, mean ± SD) returned to the previous levels after 30–35 min (10–15 min after the addition of FPL631476) and 35–40 min (15–20 min after the addition of FPL64176), whereas the RRPAs of high-KCl-induced contractions in the progesterone + vehicle groups (*n* = 6, mean ± SD) still decreased after the addition of the vehicle. Significant differences were observed between the two groups at 20–25, 25–30, 30–35, and 35–40 min. The RRPAs of high-KCl-induced contractions in the FPL64176-only group (*n* = 6, mean ± SD) increased in a time-dependent manner for 40 min, whereas the RRPAs of high-KCl-induced contractions in the vehicle-only groups (*n* = 6, mean ± SD) were stable for 40 min (Figure 4F and Appendix A). Significant differences were observed between the two groups at five-min intervals after FPL64176 was added. FPL64176 enhanced the high-KCl-induced contractions via L-type VDCC, and the recovered high-KCl-induced contractions decreased by progesterone via the L-type VDCC.

## 3. Discussion

Uterine contractions are essential for parturition, and it has recently been reported that increased intrauterine pressure during early pregnancy is crucial for fetal morphogenesis [28]. Therefore, uterine contractions play an important role not only in parturition but, also, in early fetal development. Various hormones are known to affect the morphological and functional changes in the pregnant uterus, including responses to stimuli and relaxants during pregnancy. In this study, we found that natural steroid hormones, such as estradiol, testosterone, cortisol, and aldosterone, did not cause contractions in the resting myometrium when added sequentially at a concentration of 10^−7^ M–10^−4^ M. Conversely, progesterone caused continuous and rhythmic contractions, similar to labor, at 5 × 10^−7^ M. This is the first study to demonstrate that progesterone causes contractions at concentrations as low as 5 × 10^−7^ M. The contractions increased in a concentration-dependent manner to a maximum at 5 × 10^−6^ M; however, progesterone-induced contractions began to decrease at 10^−5^ M and disappeared within 10 min at concentrations >5 × 10^−5^ M. Furthermore, we evaluated the effects of progesterone when administered in a single dose and found that progesterone caused continuous and rhythmic contractions within 10 min at 5 × 10^−7^ M and rapidly induced contractions at concentrations >10^−6^ M, which increased in a time-dependent manner, peaking at 5 × 10^−6^ M. However, at concentrations >5 × 10^−5^ M, early progesterone-induced contractions decreased within seconds and subsequently disappeared within minutes. Together, these results suggested that the resting myometrium responds to progesterone at concentrations of 5 × 10^−7^ M–10^−5^ M by contracting rhythmically but reacts to progesterone at concentrations >5 × 10^−5^ M and relaxes. 

In addition to the stimulatory effect on contractile activity, progesterone enhanced oxytocin-induced contractions at 5 × 10^−7^ M–10^−5^ M; however, no further progesterone-derived stimulation was observed beyond this, and oxytocin-induced contractions were inhibited by a progesterone concentration >5 × 10^−5^ M. Thus, progesterone appears to exert conflicting stimulatory and inhibitory effects on contractile activity in an oxytocin-contracted myometrium, and this inhibitory effect may explain why contractions decreased in the resting myometrium at a concentration >5 × 10^−5^ M. Moreover, whether the nongenomic effects of progesterone are inhibitory [13,14,17,25] or stimulatory [18,19,20,21] has been fiercely debated. Our experimental results may provide a possible solution to this debate by demonstrating that progesterone exerts conflicting nongenomic effects that vary with its concentration. Unexpectedly, progesterone did not enhance high-KCl-induced tonic contractions; however, it inhibited contractions at concentrations of 5 × 10^−7^ M–10^−4^ M. This inhibitory effect was independent of tissue toxicity, as the myometrial response to high-KCl was recovered after the solution was replaced several times. Together, these results suggest that the stimulatory effect of progesterone in the resting and oxytocin-contracted myometrium at concentrations of 5 × 10^−7^ M–10^−5^ M is abolished during high-KCl-induced contractions, allowing its inhibitory effects to be imparted at concentrations as low as 5 × 10^−7^ M. Thus, the stimulatory effect of progesterone may be dominant at concentrations of 5 × 10^−7^ M–10^−5^ M. The mechanism involved in the inhibitory effect of progesterone at high concentrations (>5 × 10^−5^ M) may be explained as follows. Progesterone may interfere with the oxytocin/oxytocin receptor axis at high concentrations and inhibit contractions, because progesterone is known to combine with the oxytocin receptor. Additionally, the oxytocin/oxytocin receptor is known to increase intracellular Ca^2+^, which activates cytosolic phospholipase A2, stimulating the formation of arachidonate from membrane glycerophospholipids. The arachidonate is converted to prostaglandin by prostaglandin endoperoxide-H synthases-1 and -2. Thus, the oxytocin/oxytocin receptor is involved in prostaglandin synthesis [29]. Therefore, the inhibitory effect of progesterone at pharmacological concentrations may, in part, depend on its effects on the oxytocin/oxytocin receptor axis and prostaglandin synthesis.

Recently, it was reported that the inhibitory effect of progesterone (10^−8^ M–10^−4^ M) on KCl-induced contractions was negligible [30]. The authors evaluated whether progesterone inhibited KCl (25 mM)-induced rhythmic contractions by using uterine ring tissue, which consists of longitudinal muscles and circular muscles, obtained from 22-day-pregnant Sprague Dawley rats. In contrast, we evaluated the inhibitory effect of progesterone on high-KCl (72.7 mM)-induced tonic contractions using myometrial strips consisting of longitudinal muscles obtained from 20-day-pregnant Wistar rats. We used myometrial strips in our experiments, because the longitudinal muscles play a principal role in the evacuation of conceptus in the late stages of pregnancy. The different results between the two studies may be due to differences in the pregnant rats, sampled tissues, and concentrations of KCl used, as well as the type of contractions.

Progesterone is known to combine with nPR in the cytosol and translocate into the nucleus, where it modulates the gene expression [31]; therefore, progesterone is generally thought to act via genomic effects over a period of hours or days. However, the conflicting effects of progesterone observed in this study are thought to be nongenomic, as they occurred within seconds or minutes. Additionally, we found that RU486 did not block progesterone-induced contractions (5 × 10^−6^ M) in the resting myometrium, early progesterone (5 × 10^−5^ M)-induced contractions, the subsequent disappearance of contractions, or the inhibitory effect of progesterone (5 × 10^−6^ M) on high-KCl-induced contractions. These findings suggest that progesterone causes rapid and rhythmic myometrial contractions and inhibits high-KCl-induced contractions via nongenomic pathways through different membrane receptors. Moreover, the nongenomic pathway involved in the stimulatory effects of progesterone may be disturbed in high-KCl-induced contractions.

A high KCl inhibits intracellular K^+^ efflux via the K^+^ channel, causing slight plasma membrane depolarization, which causes extracellular Na^+^ influx via Na^+^ channels. This Na^+^ influx then enhances plasma membrane depolarization in smooth muscle cells to primarily activate VDCCs, which cause a Ca^2+^ influx and, thus, activate calmodulin. The activated calmodulin (Ca^2+^-calmodulin) complex induces MLC phosphorylation via the MLC kinase and causes tonic contractions. Simultaneously, the oxytocin–oxytocin receptor couples with G_q/11_ proteins stimulate a phosphoinositide turnover and activate phospholipase C, thus producing inositol triphosphate (IP_3_) and diacylglycerol [32,33,34]. IP_3_ releases Ca^2+^ from intracellular stores via the IP_3_ receptor and causes a Ca^2+^ influx via VDCCs, which stimulates a further Ca^2+^ release from intracellular stores via the ryanodine receptor [35]. However, the intracellular Ca^2+^ concentration required to cause agonist-induced rhythmic contractions is lower than that in high-KCl-induced tonic contractions [36]. Diacylglycerol causes a Ca^2+^ influx via ROCCs [37,38,39] and protein kinase C activation, which inhibits the dephosphorylation of phosphorylated MLC by inactivating MLC phosphatase, thus inhibiting relaxation. After causing contractions, the intracellular Ca^2+^ levels are decreased by calcium pumps, calcium-dependent (sensitive) K^+^ channels, voltage-dependent K^+^ channels, uptake into intracellular Ca^2+^ stores, and the inhibition of Ca^2+^ influx by refractory VDCCs. Consequently, decreased intracellular Ca^2+^ levels induce a reverse reaction in contractile pathways that relaxes contractions. These periodic increases and decreases in intracellular Ca^2+^ levels are known as calcium oscillations [40,41,42]; however, this process does not occur in high-KCl-induced tonic contractions. Nevertheless, changes in the intracellular levels of cations, including Ca^2+^, and/or the contractile pattern may be related to the different responses to progesterone during oxytocin-induced rhythmic contractions and high-KCl-induced tonic contractions.

In this study, we revealed that rapid and rhythmic progesterone-induced contractions were caused by an increase in intracellular Ca^2+^ levels via plasma membrane channels, as contractions did not occur in calcium-free PSS. In addition, these plasma membrane channels may have been L-type VDCCs, because the contractions were completely blocked by the pretreatment with verapamil and nifedipine, thus indicating that the putative L-type VDCCs play an important role in the stimulatory effect of progesterone on the resting myometrium. However, the underlying mechanisms in this pathway remain unclear. A high KCl is known to cause contractions mainly via L-type VDCCs, which are also involved in oxytocin-induced contractions. Therefore, it has been hypothesized that the inhibitory effect of progesterone involves the closing of conventional L-type VDCCs, because progesterone inhibits high-KCl-induced contractions and the putative L-type VDCCs involved in progesterone-induced contractions are closed during high-KCl-induced contractions when conventional L-type VDCCs are opened. This hypothesis is further supported by the following findings: (a) FPL64176 (an activator of L-type calcium channels) increased the high-KCl-induced contractions and repressed the inhibitory effect of progesterone during high-KCl-induced contractions. (b) Progesterone, which may exert a stimulatory effect via the putative L-type VDCC, could not increase the high-KCl-induced contractions, although oxytocin, which exerts a stimulatory effect via conventional L-type VDCCs, could increase the high-KCl-induced contractions (data not shown). Additionally, our findings suggest that FPL6417 is an activator of conventional L-type VDCCs.

These results suggest that progesterone increases intracellular Ca^2+^ levels by opening putative L-type VDCCs via nongenomic progesterone/membrane receptor pathways, resulting in an increased intracellular voltage that further increases the intracellular Ca^2+^ levels by opening conventional L-type VDCCs. Moreover, high intracellular Ca^2+^ levels are decreased by the refractoriness of putative L-type VDCCs to progesterone and/or the closing of conventional L-type VDCCs via a nongenomic pathway involving progesterone/other membrane receptors. In addition, the opening of putative L-type VDCCs may be faster than the closing conventional L-type VDCCs. Recently, mPRα has been suggested as a candidate membrane receptor involved in closing conventional L-type VDCCs, because it has been reported to be involved in decreasing intracellular Ca^2+^ in vascular smooth muscles [12]. 

VDCCs consist of α_1_, α_2_, β, and γ subunits. The α_1_ subunit is encoded by 10 different Cav genes; possesses an electronic sensor and channel pores; and assists with functional classification (L-, P/Q-, N-, R-, and T-types) based on the electrophysiological and pharmacological properties [43]. The α_1_ subunits of L-type VDCCs can be subclassified as Cav1.1, Ca_v_1.2, Ca_v_1.3, or Ca_v_1.4 and are expressed in various tissues, including the skeletal muscle; heart, brain, and retina; and endocrine organs. Ca_v_1.2 is primarily expressed in smooth muscles and the myocardium and is involved in contractions. The α_2_δ, β, and γ subunits play important roles in regulating the expression, function, and intracellular localization of the α_1_ subunit [44]; therefore, their modulation may be involved in the conflicting nongenomic effects of progesterone. Thus, the subunits of the putative L-type VDCCs may be functionally and/or structurally different from those of conventional L-type VDCCs. 

However, there may be another reason for these conflicting nongenomic effects—namely, the stimulatory action caused by putative L-type VDCCs. In this study, we found that progesterone causes pregnant myometrial contractions at a concentration of 5 × 10^−7^ M via a nongenomic pathway. Consistently, recent studies have demonstrated that progesterone (5 × 10^−7^ M) increases intracellular Ca^2+^ levels in the human sperm via a nongenomic pathway and thereby enhances sperm motility by a process known as hyperactivation [45,46]. It has been reported that the increase in intracellular Ca^2+^ levels in human sperm is caused by a specific VDCC known as the sperm-specific Ca^2+^ channel (CatSper), which is located in the main section of the flagellum [47,48,49]. Additionally, progesterone can activate the orphan progesterone-dependent lipid hydrolase α/β hydrolase domain-containing protein 2, which is expressed at high levels in the sperm plasma membrane and depletes the endocannabinoid 2-arachidonoylglycerol (2AG) levels, which typically inhibits CatSper. Therefore, progesterone can prevent the inhibition of CatSper by decreasing the 2AG levels, thereby increasing the intracellular Ca^2+^ levels and inducing sperm hyperactivation [50]. These findings indicate the possibility that a myometrium-specific calcium channel exists that is involved in the stimulatory action of progesterone; however, this is unlikely to be CatSper itself, because we found that 2AG (10^−6^ M) did not inhibit progesterone (10^−6^ M)-induced contractions (data not shown). To date, no study has reported that a putative myometrium-specific calcium channel is involved in the increase of intracellular Ca^2+^ caused by progesterone in the pregnant myometrium; therefore, further studies are required to determine whether a putative myometrium-specific calcium channel is involved in mediating the stimulatory effect of progesterone on contractile activities.

In this study, we found conflicting nongenomic effects of progesterone on contractile activities in the pregnant myometrium. We showed that the stimulatory effect of progesterone was dominant at physiological concentrations and the inhibitory effect was dominant at pharmacological concentrations. However, progesterone inhibited high-KCl-induced tonic contractions at physiological concentrations. These findings lead us to believe that the stimulatory effect may be involved in the onset of parturition in rats and humans having no withdrawal of progesterone in the late stage of pregnancy and that progesterone may prevent tonic contractions like excessive labor pain. However, our interpretation may be unacceptable to some doctors who retain the conventional view that progesterone decreases contractile activities, prevents premature labor, and maintains pregnancy. Therefore, to provide additional support and understanding of the conflicting nongenomic effects of progesterone, we need to investigate the modulation of the conflicting effects of progesterone during pregnancy. It will be interesting to determine whether progesterone shows the conflicting nongenomic effects in nonpregnant, early-pregnant, middle-pregnant, and puerperium myometrium through the action of putative and conventional VDCCs. Additionally, it will be interesting to determine whether estrogen has a nongenomic inhibitory effect on contractile activities and whether it can abolish the nongenomic stimulatory effect of progesterone during pregnancy. Previously, we reported that dydrogesterone, a derivative of retroprogesterone with a reversed three-dimensional structure, exhibited a nongenomic inhibitory effect on contractile activities in the rat and human pregnant myometrium [16]. Therefore, we are interested in the effect of dydrogesterone on progesterone-induced contractions and the efficacy of dydrogesterone on preventing premature labor via nongenomic pathways. Further investigation is needed to address the issues underlying the roles and mechanisms of the conflicting nongenomic effects of progesterone.

## 4. Materials and Methods

### 4.1. Chemicals 

Progesterone, 17β-estradiol, testosterone, cortisol, aldosterone, RU486, nifedipine, verapamil, FPL6417, and dimethyl sulfoxide (DMSO) were purchased from Sigma-Aldrich (St. Louis, State of Missouri, USA). Oxytocin was obtained from ASKA Pharmaceuticals (Tokyo, Japan). All chemicals were dissolved in DMSO. 

### 4.2. Animals 

Pregnant Wistar rats were obtained from the Oriental Bioservice Corporation (Kyoto, Japan) and housed under controlled conditions (12-h light/12-h dark cycle) with water and rat chow provided ad libitum. The rats used in this study were 12 weeks of age, and this was their first pregnancy. The vaginal plug confirmation date was defined as day 0. All rats were euthanized on day 20 by CO_2_ inhalation, and the uterus was removed and used for experiments. 

### 4.3. Myometrium Preparation and Evaluation of Contractile Activity 

Myometrial strips were prepared from the dissected rat uterus, as described previously [51,52]. Each strip (width: 2 to 3 mm, length: 10–15 mm) was attached to a holder under 1 g of resting tension, equilibrated for 30 min in PSS, and then repeatedly treated with a 72.7-mM KCl solution (high-KCl) until the response stabilized. PSS contained 136.9-mM NaCl, 5.4-mM KCl, 1.5-mM CaCl_2_, 1.0-mM MgCl_2_, 23.8-mM NaHCO_3_, 5.5-mM glucose, and 0.01-mM ethylenediaminetetraacetic acid. A solution with high KCl concentration was prepared by replacing NaCl with an equimolar quantity of KCl. Calcium-free PSS was prepared without CaCl_2_. These solutions were saturated with a 95% O_2_/5% CO_2_ mixture at 37 °C (pH 7.4). After pretreatment with high KCl, a steroid hormone or a vehicle (DMSO) was added, and the myometrial contractile activity was recorded isometrically using a force-displacement transducer (TB611T; Nihon Kohden, Tokyo, Japan) connected to a Model 3134 strain amplifier. Uterine strips in each experiment were prepared from the uterine tissue of one animal (one strip/one group), and the experiment was repeated six times on each individual rat. Data were analyzed using the Unique Acquisition software package (Microsoft Windows 7; Unique Medical, Tokyo, Japan). 

### 4.4. Effects of Progesterone and Other Steroid Hormones on the Resting Myometrium 

Myometrial strips were pretreated with high KCl for 10 min, and the peak area of contractions was used as the reference area. Five minutes after high KCl was replaced with PSS, steroid hormones were added. First, we evaluated the effect of progesterone at 10^−7^ M and then increased the concentration to 5 × 10^−7^, 10^−6^, 5 × 10^−6^, 10^−5^, 5 × 10^−5^, and 10^−4^ M every 10 min. We also evaluated the effect of other steroid hormones, such as estradiol, testosterone, cortisol, and aldosterone, on the myometrium at the same concentrations as progesterone. The RRPA (steroid hormone-induced peak area for 10 min/KCl-induced peak area for 10 min) was compared between the steroid hormone and vehicle groups. We evaluated the effects of a single application of progesterone on the myometrial response. Myometrial strips were pretreated with high KCl for 5 min, and the peak area was defined as the reference area. Five minutes after high KCl was replaced with PSS, progesterone was added (10^−7^, 5 × 10^−7^, 10^−6^, 5 × 10^−6^, 10^−5^, 5 × 10^−5^, or 10^−4^ M). The myometrial response to progesterone at each concentration was observed for 20 min, and the contractile activity was evaluated using the progesterone-induced peak area/KCl-induced peak area for 5 min at 5-min intervals (−5–0, 0–5, 5–10, 10–15, and 15–20 min). 

### 4.5. Effects of Progesterone on Oxytocin- and High-KCl-Induced Myometrial Contractions

Myometrial strips were pretreated with high KCl for 10 min, and the peak area was defined as the reference area. Five minutes after high KCl had been replaced with PSS, contractions were induced using oxytocin (100 μunit/mL) or high KCl and observed for 10 min. Progesterone was then added to the PSS and high KCl, and the concentrations were increased sequentially every 10 min from 10^−7^ M to 5 × 10^−7^, 10^−6^, 5 × 10^−6^, 10^−5^, 5 × 10^−5^, and 10^−4^ M. An equivalent volume of DMSO was added to the vehicle group at the same time as progesterone. The RRPA was compared between the progesterone- and vehicle-treated groups. 

### 4.6. Effects of RU486 on Progesterone Action

Myometrial strips were preloaded with high KCl for 5 min, and the peak area was defined as the reference area. Quiescent myometrial strips were pretreated for 60 min with RU486 at 10^−7^ M, because this concentration has been reported to be sufficient for binding with nPR, with a K_d_ value of 3.6 x 10^−9^ M [53]. Progesterone (5 × 10^−6^ or 5 × 10^−5^ M) was then added, and the effect of RU486 on the action of progesterone was observed for 20 min. Contractions were also induced in the RU486-pretreated myometrial strips using high KCl to evaluate the effect of RU486 on the action of progesterone (5 × 10^−6^ M) during agonist-induced contractions for 20 min. In the vehicle group, myometrial strips were pretreated with an equivalent volume of DMSO for 60 min. The effect of RU486 on the action of progesterone was evaluated using the relative RRPA at 5-min intervals (−5–0, 0–5, 5–10, 10–15, and 15–20 min). The RRPA was compared between the RU486 and vehicle groups.

### 4.7. Effects of Calcium-Free PSS and Calcium Channel Inhibitors on Progesterone in the Quiescent Myometrium

Myometrial strips were preloaded with high KCl for 5 min, and the peak area was used as the reference area. Five minutes after high KCl was replaced with PSS and calcium-free PSS, progesterone (5 × 10^−6^ M) was added to each solution, and the contractile response was observed for 20 min. Similarly, myometrial strips were pretreated with verapamil (5 × 10^−7^ M) and nifedipine (10^−7^ M) for 15 min in PSS before progesterone (5 × 10^−6^ M) was added, and the contractile response was observed for 20 min and evaluated using the RRPA at 5-min intervals (−5–0, 0–5, 5–10, 10–15, and 15–20 min). 

### 4.8. Effects of Calcium Channel Activator on Progesterone during High-KCl-Induced Contractions

The myometrial strips were divided into four groups: progesterone + FPL64176, progesterone + vehicle, vehicle alone, and FPL64176 alone. The myometrial strips were preloaded with high KCl for 5 min, and the peak area was used as the reference area. Five minutes after high KCl was replaced with PSS, contractions were induced in the myometrial strips using high KCl. Progesterone (5 × 10^−6^ M), FPL64176 (10^−7^ M), and DMSO were added to the appropriate groups, and the contractions were observed for 20 min. The RRPA of high-KCl-induced contractions were evaluated in all groups at more-than 5-min intervals (−5–0, 0–5, 5–10, 10–15, 15–20, 20–25, 25–30, 30–35, and 35–40 min). 

### 4.9. Statistical Analysis

Data were expressed as the mean ± standard deviation (SD) and were analyzed using JMP version 10.0.2 (SAS Institute, Cary, NC, USA). The statistical significance of differences in the measured parameters across different groups was assessed using the Steel–Dwass test for multiple comparisons or the Wilcoxon rank-sum test. Results were considered statistically significant at *p*-values <0.05.

## 5. Conclusions

We conclude that progesterone exerts conflicting nongenomic effects on the pregnant the myometrium, with stimulatory and inhibitory effects on contractile activities. Additionally, we conclude that the stimulatory effect occurs faster and is more significant than the inhibitory effect in the resting myometrium. However, the dominant stimulatory effect is abolished by a high KCl, and the recessive inhibitory effect is highlighted. The stimulatory effect may be caused via the putative L-type VDCC or myometrium-specific VDCC, whereas the inhibitory effect may be caused via the conventional L-type VDCC that is involved in high-KCl- and oxytocin-induced contractions. An improved understanding of the nongenomic stimulatory and inhibitory effects of progesterone, and further studies on the mechanisms of the progesterone–membrane receptor–calcium channel axis could help modulate parturition and treat preterm birth.

## Figures and Tables

**Figure 1 ijms-22-02154-f001:**
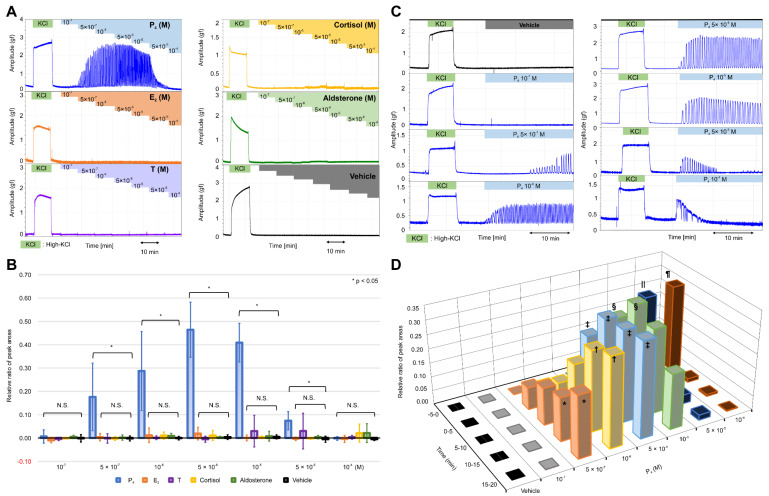
Effects of various natural steroid hormones on the contractile activities in the resting myometrium. (**A**) Representative myograph of six myographs obtained from experiments that used myometrial strips prepared from the uterine tissues of rats at 20 days of gestation. Progesterone (P_4_), 17β-estradiol (E_2_), testosterone (T), cortisol, and aldosterone were added sequentially to a physiological saline solution (PSS) at concentrations of 10^−7^ M–10^−4^ M. (**B**) The relative ratio of the peak area (RRPA) of the progesterone (P_4_), 17β-estradiol (E_2_), testosterone (T), cortisol, aldosterone, and vehicle groups in the resting myometrium. The peak area of high-KCl-induced contractions for 10 min was defined as the reference area. (**C**) Representative myograph of six myographs obtained from experiments that evaluated the effects of single-dose progesterone on the contractile activities in the resting myometrium. Progesterone was added to the PSS at the concentrations of 10^−7^–10^−4^ M. (**D**) Time- and concentration-dependent changes in the RRPA of the vehicle and progesterone groups at various progesterone concentrations. The peak area of high-KCl-induced contractions for 5 min was defined as the reference area. * *p* < 0.05, ^†^
*p* < 0.05, ^‡^
*p* < 0.05, ^§^
*p* < 0.05, ^‖^
*p* < 0.05, ^¶^
*p* < 0.05.

**Figure 2 ijms-22-02154-f002:**
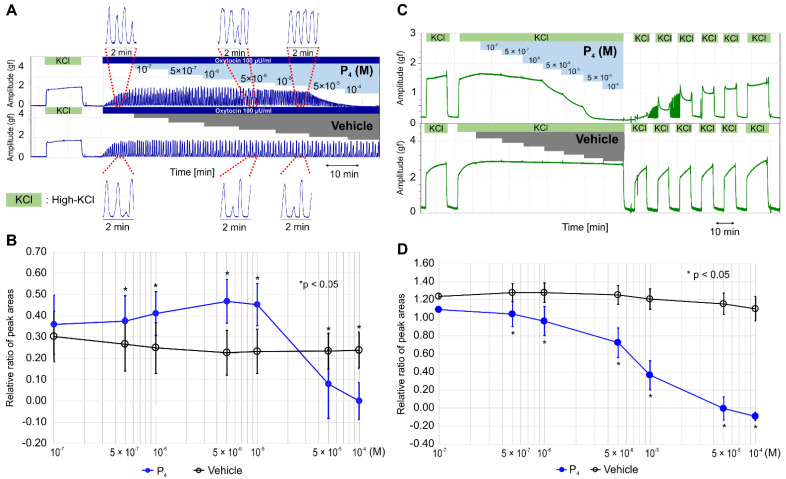
Effects of progesterone on oxytocin- and high-KCl-induced contractions. (**A**) Representative myograph of six myographs obtained when rhythmic contractions were induced by oxytocin (100 μunit/mL), and progesterone was sequentially added to a physiological saline solution (PSS) at the concentration of 10^−7^ M–10^−4^ M every 10 min after a contraction. (**B**) Time-dependent changes in the relative ratio of the peak area (RRPA) of oxytocin-induced contractions in the progesterone and vehicle groups. (**C**) Representative myograph of six myographs obtained when tonic contractions were induced by high KCl (72.7 mM), and progesterone was sequentially added into the PSS at the concentrations of 10^−7^ M–10^−4^ M every 10 min after a contraction. (**D**) Time-dependent changes in the RRPA of high-KCl-induced contractions in the progesterone and vehicle groups.

**Figure 3 ijms-22-02154-f003:**
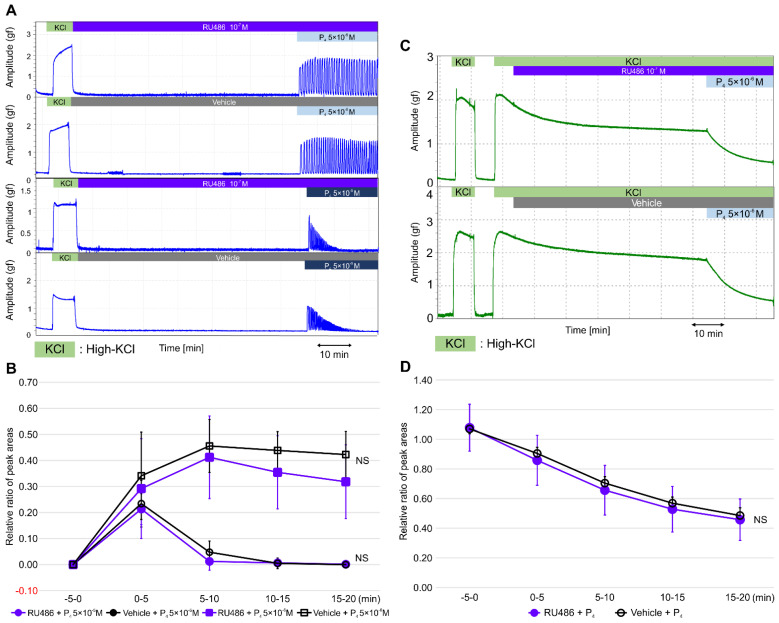
Effects of RU486 on progesterone-induced contractions in the resting myometrium and on progesterone-induced inhibitions against high-KCl-induced contractions. (**A**) Representative myograph of six myographs was obtained when the resting myometrial strips were pretreated with RU486 (10^−7^ M) or a vehicle for 60 min, progesterone was added at 5 × 10^−6^ or 5 × 10^−5^ M, and myometrial strips were cotreated with RU486 or a vehicle for 20 min. (**B**) Time-dependent changes in the relative ratio of the peak area (RRPA) of progesterone-induced contractions in the RU486- and vehicle-treated groups. (**C**) Representative myograph of six myographs obtained when myometrial strips contracted by high KCl were pretreated with RU486 or a vehicle for 60 min, progesterone was added at 5 × 10^−6^ M, and the strips were cotreated with RU486 or a vehicle for 20 min. (**D**) Time-dependent changes in the RRPA of high-KCl-induced contractions in the RU486- and vehicle-treated progesterone groups.

**Figure 4 ijms-22-02154-f004:**
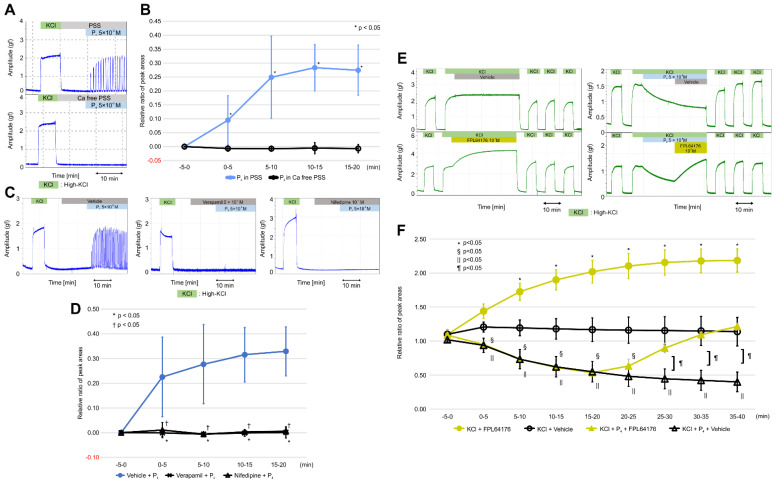
Effects of calcium-free physiological saline solution (PSS), verapamil, and nifedipine on progesterone-induced contractions in the resting myometrium, and the effect of FPL64176 on progesterone-induced inhibition against high-KCl-induced contractions. (**A**) Representative myograph of six myographs obtained when progesterone (5 × 10^−6^ M) was added into calcium-free PSS and PSS. (**B**) Time-dependent changes in the relative ratio of the peak area (RRPA) of progesterone-induced contractions in calcium-free PSS and PSS. (**C**) Effects of verapamil and nifedipine on progesterone-induced contractions, as shown by the representative myograph of six myographs obtained from experiments in which myometrial strips were pretreated for 15 min with verapamil (5 × 10^−7^ M) or nifedipine (10^−7^ M), progesterone (5 × 10^−6^ M) was added, and the strips were cotreated for 20 min with verapamil or nifedipine. (**D**) Time-dependent changes in the RRPA of progesterone-induced contractions in the verapamil- and nifedipine-pretreated groups. (**E**) Representative myograph upon progesterone (5 × 10^−6^ M), FPL6417 (10^−7^ M), or the vehicle addition in high-KCl-induced contractions. Progesterone was added 5 min after the tonic contraction was induced by high KCl. FPL64176 or the vehicle was added 20 min after the addition of progesterone. (**F**) Time-dependent changes in the RRPA of high-KCl-induced contractions in the FPL64176 alone, vehicle alone, progesterone-FPL64176, and progesterone-vehicle groups.

## Data Availability

All data are included in the article.

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
