# Peer review of "Conflicting Nongenomic Effects of Progesterone in the Myometrium of Pregnant Rats"

_ijms, 2021, doi:10.3390/ijms22042154_

Round 1

Reviewer 1 Report

Yasuda and colleagues have written a well-balanced original research paper, highlighting the fact that progesterone may affect the contractile activity of pregnant myometrium via non-genomic pathways, by performing in vitro experiments using myometrial strips from pregnant rats. Their study revealed that progesterone caused myometrial contraction in a concentration- and time-dependent manner at concentrations up to 5*10-7 M, while this effect decreased at higher concentrations.

The flow of work that the authors want to communicate as well as the main ideas seem easy enough to follow for the reader. The Materials and Methods section is concise yet clear, making the experiment reproducible, while the results support the stated hypothesis. Figures seem free from apparent manipulation and are of high enough quality for the reader to interpret.

The subject is of interest, and, as the authors have highlighted, no study has reported the involvement of a putative myometrium-specific calcium channel in the increase of intracellular Ca2+ caused by progesterone in the pregnant myometrium, thus making the authors’ contribution to the field that much more valuable. With this positive outlook in mind, I do have some comments that I believe the authors should address in order to improve their paper.

For instance, the authors state that “Previously, progesterone was thought to exert specific effects on target organs via genomic pathways through the nuclear progesterone receptor (nPR); however, recent evidence has suggested that progesterone acts via non-genomic pathways [13–21].” – perhaps they could briefly mention the mechanism involved in the genomic pathways, so ar to better highlight the difference between these pathways.

Further on, they also mention that “Although progesterone’s role in decreasing uterine contraction via genomic pathways has been studied extensively, few studies have investigated whether steroid hormones, including progesterone, affect the contractile activity of pregnant myometrium via non-genomic pathways” – a couple of such examples should perhaps be mentioned.

I have no hesitation in saying that the writing is generally very clear and well organized, and I have found no issues regarding this aspect.

Ultimately, I would ask the authors to perhaps concentrate a bit more on the clinical aspects and applicability of this research. While they do mention that further studies are required to determine the involvement of myometrium-specific calcium channels in mediating the effects of progesterone and that an improved understanding of the non-genomic effects of progesterone could help modulate parturition and treat preterm birth, they could be a bit more specific by suggesting potential therapeutic approaches.

Overall, I find this paper of high enough quality to be accepted for publication after minor changes.

Author Response

Please see it in the attachment.

Reviewer 2 Report

Brief summary:

The aim of this study was to evaluate the non-genomic effect of P4 stimulation on the contractile activity of pregnant rat myometrium. In general, P4 stimulated the contractile activity of the myometrium when used at smaller concentrations, but the increased concentration of P4 (10-5, 10-4) decreased the contractile activity of the myometrium. The experiments indicate that the observed P4-dependent effects are non-genomic via putative L-type voltage-dependent calcium channels.

General comment:

I find this article interesting, well-written, with valuable content. The introduction provides sufficient background of the study and the aim of the study is clear. I believe that the fact that the Authors have incorporated some background into the results section is also an advantage, as it allows the reader to easily follow the presented content. Nevertheless, it possesses some flaws - please see them specified below. The discussion is nice to follow, however, it could be improved (please see the suggestions below). The materials and methods section is fine, however, some information could be added (please find suggestions below). The conclusions are supported by the results.

Specific comments:

Results section:

1/ in my opinion, the provided figures are too small and are hardly readable. I recommend increasing the size of the font and/or increasing the size of each panel. In the current form I could read them only when opening electronically with a big zoom, and carefully following the description of the results. Otherwise, ex. in a printed version they are unreadable.

2/ paragraph 2.2. and 2.3 are the same. Please remove one of them.

Discussion section:

The conflicting effect of P4 action on contractile activity might be also coupled with P4-dependent alterations in oxytocin receptor concentration in the tissue and the effect of P4 on prostaglandins synthesis and release from the myometrium. Please consider adding a paragraph discussing these relations with the obtained within this study results. 

These of my concerns raised since in lines 71-74 the Authors present the concentration of P4 in the blood plasma of women and rats. The Authors indicate, that in rats its concentration increase to 5 x 10 -7 M during pregnancy. In the study, also a higher concentration is used (up to 10-4 M). As the results show, the physiological concentrations of P4 increased the contractile activity of the myometrium, and only the over-concentration of P4 contributed to the decrease of tissue contractile activity. Therefore, the used over-concentration of P4 could impact the expression of oxytocin receptors and/or prostaglandins synthesis and release, and thus, the "conflicting" effect of P4 could be visible.

Material and methods section:

1/ Please consider adding more information about the animals. Specifically, the age of the animals, details concerning the conception, information whether it was the first or the next pregnancy.

2/ Please add the number of animals. It is written in the results section, however, it would be nice to add this information also here.

Author Response

Please see it in the attachment.

Reviewer 3 Report

The manuscript IJMS-1103000 is about non-genomic action of progesterone on pregnant rat myometrium along with other steroid hormones (oestrogen, androgen, etc.). The authors found that the non-genomic action of progesterone is confusing on myometrial contractions induced by different agents, and they suppose that calcium channels may have a role in this contraction increasing or decreasing effect.

Major concerns:

  1. A recent paper deals with this issue and coming to different conclusions (https://doi.org/10.1016/j.lfs.2020.118584). Authors must their result in the light of this paper.
  2. All the experiments were done in isolated organ bath, the conclusions are not supported by other results (e.g. second messengers, membrane receptor, G-protein activation)
  3. The duration of the experiments was longer than 1 hour increasing the risk of genomic actions as well. Although the authors think that the inefficacy of mifepristone confirm the non-genomic reaction of progesterone, it is not enough for proving.
  4. The responses of the uterine samples must be plotted on a normal concentration-response curves in a semilogarithmic system to get real information about the responses.
  5. The application of high concentration of KCl (72.7mM) at the very beginning of the experiment must be explained. Such a high concentration usually added at the end of the experiment to check the viability of the tissue, it can increase the risk of tissue fatigue and may modify the response at all. Additionally, the selection of this strange 72.7 mM concentration seems rather the result of a miscalculation (maybe 70mM was the aim) than a well-defined decision…
  6. The subsequent administration of KCl and oxytocin also raises questions: why these 2 contracting agents must be given within the same experiment before adding progesterone?

Minor concern

  1. If the endometrium of uterine samples were removed before the experiment, in that case it was myometrium, but without denuding it must be called uterus.
  2. Based on the point above, the “uterine myometrium (row 34) is an inappropriate expression.

Author Response

Please see it in the attachment.

Round 2

Reviewer 3 Report

The manuscript has been improved significantly, the problematic parts have been clarified and modified.